# Exercise Normalized the Hippocampal Renin-Angiotensin System and Restored Spatial Memory Function, Neurogenesis, and Blood-Brain Barrier Permeability in the 2K1C-Hypertensive Mouse

**DOI:** 10.3390/ijms23105531

**Published:** 2022-05-16

**Authors:** Ying-Shuang Chang, Chih-Lung Lin, Chu-Wan Lee, Han-Chen Lin, Yi-Ting Wu, Yao-Hsiang Shih

**Affiliations:** 1Department of Anatomy, School of Medicine, College of Medicine, Kaohsiung Medical University, 100, Shih-Chuan 1st Road, Sanmin District, Kaohsiung 80708, Taiwan; yschang0708@gmail.com (Y.-S.C.); hanchen@kmu.edu.tw (H.-C.L.); 2Graduate Institute of Medicine, College of Medicine, Kaohsiung Medical University, 100, Shih-Chuan 1st Road, Sanmin District, Kaohsiung 80708, Taiwan; chihlung1@yahoo.com; 3Department of Neurosurgery, Kaohsiung Medical University Hospital, 100, Tzyou 1st Road, Sanmin District, Kaohsiung 80756, Taiwan; 4Department of Nursing, National Tainan Junior College of Nursing, 78, Section 2, Minzu Road, West Central District, Tainan 70043, Taiwan; chuwan2009@gmail.com; 5Department of Medical Research, Kaohsiung Medical University Hospital, 100, Tzyou 1st Road, Sanmin District, Kaohsiung 80756, Taiwan; 6Department of Nursing, Tzu Hui Institute of Technology, Pingtung County 92641, Taiwan; wuyiting1224@gmail.com

**Keywords:** hypertension, 2-kidney,1-clip, renin-angiotensin system, exercise, neurogenesis, blood-brain barrier

## Abstract

Hypertension is associated with blood-brain barrier alteration and brain function decline. Previously, we established the 2-kidney,1-clip (2K1C) hypertensive mice model by renin-angiotensin system (RAS) stimulating. We found that 2K1C-induced hypertension would impair hippocampus-related memory function and decrease adult hippocampal neurogenesis. Even though large studies have investigated the mechanism of hypertension affecting brain function, there remains a lack of efficient ways to halt this vicious effect. The previous study indicated that running exercise ameliorates neurogenesis and spatial memory function in aging mice. Moreover, studies showed that exercise could normalize RAS activity, which might be associated with neurogenesis impairment. Thus, we hypothesize that running exercise could ameliorate neurogenesis and spatial memory function impairment in the 2K1C-hypertension mice. In this study, we performed 2K1C surgery on eight-weeks-old C57BL/6 mice and put them on treadmill exercise one month after the surgery. The results indicate that running exercise improves the spatial memory and neurogenesis impairment of the 2K1C-mice. Moreover, running exercise normalized the activated RAS and blood-brain barrier leakage of the hippocampus, although the blood pressure was not decreased. In conclusion, running exercise could halt hypertension-induced brain impairment through RAS normalization.

## 1. Introduction

Hypertension is one of the critical health problems in modern society. Its prevalence is expected to approach 1.5 billion people worldwide by 2025 [1]. Hypertension is a risk factor for brain structure and function impairment. In animal studies, learning and memory are impaired in mice that received the angiotensin II (Ang II) infusion [2]; The spontaneously hypertensive rat (SHR) also showed that the volume and synapse densities of hippocampal neurons were decreased [3,4]. Furthermore, evidence indicates that mid-life hypertension is associated with late-life cognitive decline [5,6,7,8]. Another study showed that mean arterial blood pressure was inverse-related to the learning and memory function [9]. The risk of cognitive decline was decreased in hypertension patients using anti-hypertension drugs such as β-blocker [10], angiotensin-converting enzyme inhibitors [11], and calcium channel blockers [12]. In view of these findings, how to prevent hypertension-associated cognitive decline becomes an important issue.

Adult hippocampal neurogenesis (AHN) is a process for neuronal cells generating from neuronal stem cells and neuronal progenitor cells. The process involves cell death [13] and survival [14]. It is been well known that AHN is related to brain function, including hippocampus-related memory formation, exhibition, and clearance [15,16,17,18]. The mentioned hippocampus function can be impaired by AHN impairment [19]. Several studies indicated that hypertension reduced the number of newborn neurons and neuronal progenitor cells in SHR and deoxycorticosterone-salt-induced hypertension in rats [20,21,22]. We have established the 2-Kidney, 1-clip (2K1C) hypertension mice model, and we found that 2K1C-hypertension impaired the hippocampal-related spatial memory, dendritic arborization, and AHN of the mice inconsistent with other hypertension models [23]. Thus, 2K1C-hypertension may impair the mouse brain function by AHN impairment.

It is well known that exercise benefits brain structure and function. Studies revealed that treadmill exercise increased the hippocampus volume and enhanced the cognitive function in preadolescent children (9–10 years of age) [24] and the elderly (59–81 years of age) [25]. Moreover, aerobic exercise increased the dentate gyrus cerebral blood flow volume in adults (21–45 years of age), which correlated to brain function [26]. In animal studies, the evidence demonstrated that the voluntary running wheel enhances hippocampal-related structures, including volume [26], vascular density [27], synaptic plasticity [28], AHN [26,27,28], and memory function [27,28] in mice. Moreover, moderate treadmill exercising can enhance the AHN and improve the Alzheimer’s disease transgenic mice brain function [29,30]. Hence, exercise may be a possible mechanism to restore hypertension-induced AHN and brain function impairment.

Since our 2K1C-hypertension model demonstrated the AHN impairment and hippocampus-related memory function [23], the activated renin-angiotensin system (RAS) has become a potential target related to AHN impairment [31]. Other studies showed that the administration of the angiotensin II receptor blocker enhances AHN and angiogenesis through the BDNF-related pathway [32,33]. Thus, we hypothesize that 2K1C-hypertension impaired the mouse brain function by AHN impairment caused by RAS activation.

To elucidate whether exercising restored the 2K1C-hypertension-induced brain function and AHN impairment, the 2K1C-hypertension mice would perform the treadmill exercise. First, we found that exercising restored the 2K1C-hypertension mouse spatial memory function and reversed the AHN impairment. Then, the treadmill exercise restored the induced brain RAS activation and the hippocampus blood-brain barrier (BBB) leakage in the 2K1C-hypertension mouse model. Furthermore, the treadmill exercise restored the 2K1C-hypertension-induced brain RAS activation.

## 2. Results

### 2.1. Treadmill Exercise Improved the Hypertension-Induced Spatial Memory Impairment

To investigate whether treadmill exercise can ameliorate hypertension-induced spatial memory impairment, we performed a five-week treadmill exercise on the mice after 2K1C surgery one month later. We then examined the hippocampus-related spatial memory in mice by the Morris water maze behavior test. In the training phase, the escape latency time of the 2K1C group was significantly increased compared with the sham group (Figure 1, repeated two-way ANOVA, time factor, F = 9.838, df 2/38, *p* = 0.0004; 2K1C factor, F = 9.440, df 1/19, *p* = 0.0063). Notably, the escape latency time of the 2K1C with the exercise group (2K1C-EX) was significantly decreased compared with the 2K1C group (Figure 1, repeated two-way ANOVA, time factor, F = 13.46, df 2/30, *p* < 0.0001; 2K1C factor, F = 18.16, df 1/15, *p* = 0.0007). In the probe test, with the escape platform removed, the 2K1C group mice spent less time than the sham, 2K1C-EX, and sham with the exercise group (Sham-EX) (Figure 1, Ordinary one-way ANOVA, Holm-Sidak’s multiple comparisons, Sham vs. 2K1C: *p* < 0.0001; 2K1C-EX vs. 2K1C: *p* < 0.0001; Sham-EX vs. 2K1C: *p* = 0.0006). These results indicated that treadmill exercise restores hypertension-induced spatial memory impairment.

### 2.2. Treadmill Exercise Prevents the 2K1C-Induced Adult Hippocampal Neurogenesis Impairment

To investigate whether treadmill exercise can prevent the 2K1C-induced AHN impairment, we performed immunohistochemistry to observe the number of hippocampal neural stem cells (Ki67^+^ cell) and neuronal progenitor cells (DCX^+^ cell). The results showed that treadmill exercise could prevent the 2K1C-induced neuronal stem cells and neuronal progenitor cells from decreasing compared with the sham group (Figure 2A, Ki67, Ordinary one-way ANOVA, Holm-Sidak’s multiple comparisons, Sham vs. 2K1C: *p* = 0.0383; 2K1C vs. Sham-EX: *p* = 0.0330; 2K1C vs. 2K1C-EX: *p* = 0.0230; Figure 2B, DCX, Sham vs. 2K1C: *p* = 0.0188; 2K1C vs. Sham-EX: *p* = 0.0119; 2K1C vs. 2K1C-EX: *p* = 0.0430). Moreover, we found that dendritic complexity was significantly decreased in the 2K1C hypertensive mouse model (Appendix A, Repeated two-way ANOVA, radius factor x treat factor, F = 3.631, df 50/250, *p* < 0.0001). Furthermore, the treadmill exercise could restore the dendritic complexity (Appendix A, Repeated two-way ANOVA, radius factor x treat factor, F = 2.855, df 50/300, *p* < 0.0001).

Since a previous study showed that Ang II could lead to mouse hippocampal neuronal cell apoptosis [34], we intend to study whether treadmill exercise could preserve the neuron cell under 2K1C-induced hypertension. As previously described, we evaluated the pyknotic neuronal cell numbers from CA1 and the dentate gyrus (DG) by H&E staining [35]. The results showed that treadmill exercise could prevent the 2K1C-induced neuronal cell pyknosis compared with the sham group (Figure 3, CA1, Ordinary one-way ANOVA, Holm-Sidak’s multiple comparisons, Sham vs. 2K1C: *p* = 0.0001; 2K1C vs. Sham-EX: *p* = 0.0068; 2K1C vs. 2K1C-EX: *p* = 0.0226; DG, Sham vs. 2K1C: *p* = 0.0004; 2K1C vs. Sham-EX: *p* = 0.0021; 2K1C vs. 2K1C-EX: *p* = 0.0040). Furthermore, we found that treadmill exercise could prevent the 2K1C-induced neuronal cell apoptosis of the dentate gyrus (Appendix A, Ordinary one-way ANOVA, Holm-Sidak’s multiple comparisons, Sham vs. 2K1C: *p* < 0.0001; 2K1C-EX vs. 2K1C: *p* < 0.0001; Sham-EX vs. 2K1C: *p* < 0.0001). In taking all these results together, treadmill exercise could prevent the 2K1C-hypertension that impaired the mice AHN and hippocampal neuron pyknosis.

### 2.3. Treadmill Exercise Restores the Angiotensin II Level without Improving the Elevated Blood Pressure

It has been shown that 2K1C induced hypertension via RAS stimulation [31,36]. Meanwhile, the activated RAS could impair spatial memory function [2]. Therefore, we assumed that treadmill exercise improved the hypertension mouse spatial memory function via restoring the RAS. We collected the mouse plasma after the water maze task finished one day later and measured the Ang II level with a commercial ELISA kit as mentioned in the methods paragraph. The ELISA results showed that treadmill exercise could restore the elevated Ang II in the 2K1C-hypertensive mice (Figure 4A, Ordinary one-way ANOVA, Holm-Sidak’s multiple comparisons, Sham vs. 2K1C: *p* < 0.0001; 2K1C vs. Sham-EX: *p* < 0.0001; 2K1C vs. 2K1C-EX: *p* = 0.0001).

Meanwhile, unexpectedly, we found that treadmill exercise did not improve 2K1C-induced hypertension (Figure 4B, Repeated two-way ANOVA, Holm-Sidak’s multiple comparisons, SBP: systolic blood pressure, Twenty-one days after surgery, Sham vs. 2K1C: *p* < 0.0001; Sham vs. 2K1C-EX: *p* = 0.0157; 2K1C vs. Sham-EX: *p* < 0.0001; Sham-Ex vs. 2K1C-EX: *p* = 0.0002. After the exercise trial, Sham vs. 2K1C: *p* = 0.0001; Sham vs. 2K1C-EX: *p* = 0.0006; 2K1C vs. Sham-EX: *p* < 0.0001; Sham-Ex vs. 2K1C-EX: *p* < 0.0001). Moreover, we found that blood pressure was correlated with IgG2a leakage levels in different hippocampus regions (Appendix A.). These results indicated that treadmill exercise could decrease the 2K1C-induced RAS activation without improving hypertension.

### 2.4. Treadmill Exercise Decreases the Angiotensin II and Angiotensin II Receptor Type I Level of the Hippocampus in the Brains of Hypertension Mice

The association between RAS and AHN has been studied for some time. Most believed that Ang II and Ang II-related mechanisms would impair neurogenesis [32,33]. Hence, we intend to study whether treadmill exercise could restore the hippocampal RAS. We examined the level of Ang II, angiotensin II receptor type I (AGTR1), and angiotensin II receptor type II (AGTR2) from the 2K1C-hypertensive mouse model after the behavior test by immunoblotting. We found that the 2K1C-hypertension significantly increased Ang II and AGTR1 in the hippocampus, and the treadmill exercise could counteract the 2K1C-hypertensive effect. (Figure 5, Ordinary one-way ANOVA, Holm-Sidak’s multiple comparisons, Ang II: Sham vs. 2K1C: *p* = 0.0003; 2K1C vs. Sham-EX: *p* = 0.0018; 2K1C vs. 2K1C-EX: *p* = 0.0016; AGTR1: Sham vs. 2K1C: *p* < 0.0001; 2K1C vs. Sham-EX: *p* < 0.0001; 2K1C vs. 2K1C-EX: *p* = 0.0003). In addition, neither of the 2K1C-hypertension and treadmill exercises did affect the AGTR2 expression level in the hippocampus.

### 2.5. Treadmill Exercise Prevents the Blood-Brain Barrier Leakage of the Hippocampus in the Brains of Hypertension Mice

Next, we intend to investigate how treadmill exercise improves AHN and spatial memory function. A previous study had shown that Ang II would impair the BBB through endothelium cell AGTR1-related oxidative stress production [37], and another study showed that hippocampal BBB leakage would lead to neurogenesis impairment [38]. Moreover, previous studies proved that exercise could attenuate BBB disruption in different experimental models [39,40]. Therefore, we assume that the treadmill exercise might improve AHN and spatial memory function via BBB integrity restoration.

We applied the mouse IgG antibody staining to determine the BBB integrity of 2K1C-hypertensive mice as previously described [41]. We found that treadmill exercise decreased the IgG leakage level of the hippocampus in the brain of hypertensive mice (Figure 6A,B, Ordinary one-way ANOVA, Holm-Sidak’s multiple comparisons, CA1, Sham vs. 2K1C: *p* < 0.0001; 2K1C vs. Sham-EX: *p* < 0.0001; 2K1C vs. 2K1C-EX: *p* = 0.0097; DG, Sham vs. 2K1C: *p* < 0.0001; 2K1C vs. Sham-EX: *p* < 0.0001; 2K1C vs. 2K1C-EX: *p* = 0.0020). Moreover, we measured the plasma S100B protein level by ELISA to evaluate the degree of BBB disrupted in our animal model as previously described [42]. Surprisingly, we found that treadmill exercise did not decrease the hypertension plasma S100B protein level of the mice (Figure 6C, Ordinary one-way ANOVA, Holm-Sidak’s multiple comparisons, Sham vs. 2K1C: *p* = 0.0267; Sham vs. 2K1C-EX: *p* = 0.0364; 2K1C vs. Sham-EX: *p* = 0.0043; Sham-EX vs. 2K1C-EX: *p* = 0.0068). These results implied that treadmill exercise might ameliorate 2K1C-induced BBB disruption.

## 3. Discussion

Our findings are consistent with a previous study that found that exercise can decrease plasma Ang II [43]. We further found that treadmill exercise could reduce the Ang II and ATGR1 in the hippocampus of 2K1C-hypertensive mice. The mechanism of how exercising normalizes the hippocampal RAS remains unclear. A heart failure-related animal study indicated that exercise could normalize several angiotensin-converting enzyme I/II brain region ratios, not only in the rostral ventrolateral medulla region [44]. Furthermore, previous studies showed that Ang II could increase brain blood vessel reactive oxidative stress, induce inflammation, and decrease cerebral blood flow [45,46], and Ang II treatment could induce neuroinflammation via AGTR1-related signal activation and cause astrocyte senescence by excessive reactive oxidative stress production [47,48]. Therefore, in this study, we assume that exercise could improve the spatial memory function by restoring the hippocampal RAS in the 2K1C-hypertension mice.

Interestingly, we found that exercise would not decrease systolic blood pressure in our 2K1C-hypertension model. It has been well known that exercise can help control hypertension in humans [49,50], and the possible mechanism might downregulate the neuronal inflammation and activation of the paraventricular nucleus (PVN) of the hypothalamus [43]. However, an animal study showed that exercise decreases the blood pressure of SHR, but there was a significant difference between the SHR-exercise and Wistar Kyoto rats control group blood pressure [43]. The similar and confusing results that exercise did not significantly control blood pressure seem very common in hypertensive animal studies, even though exercise could normalize the RAS, as in our findings [51,52,53]. The mechanism of how exercise only decreased the 2K1C-hypertension mice plasma Ang II but did not affect blood pressure remains unclear. In conclusion, the difference between human and animal studies about how exercise affects blood pressure needs further study to fill the gap.

Besides the difference between humans and animals, the complexity of how exercise controls blood pressure also needs to be considered. For example, exercise could downregulate the blood pressure by the angiotensin (1–7) pathway under normal conditions [54,55,56], but exercise instead elevates blood pressure with RAS hyperactivation [57]. Moreover, exercising might be unable to lower blood pressure via an AGTR2-related mechanism due to the vasculature nitric oxide synthase impairment [58]. It is worthy of note that the exercise benefit seems to not only correlate to blood pressure changes. A pharmaceutical study showed that the ATGR1 antagonist can improve the BBB leakage without lowering the blood pressure in the spontaneously hypertensive rat model [59]. From the above evidence and this study, the exercise benefit is associated with the RAS normalizing instead of blood pressure.

It remains unclear how exercise improves hypertension-induced BBB leakage. So far, most studies have focused on the exercise effect on hypothalamic BBB leakage. For example, an animal study showed that exercise could maintain the BBB integrity and decrease leakage in the PVN, nucleus of the solitary tract, and rostral ventrolateral medulla, which might help blood pressure control [60]. The PVN regulates the adrenocorticotropic hormone and antidiuretic hormone-releasing, which are essential in blood pressure homeostasis maintenance [61]. Unlike the other brain areas, the capillaries of PVN lack the endothelial cell tight junction-which called fenestrated capillaries [62,63]. This unique structure allows polypeptides to enter the PVN without changing BBB and make the PVN maintain the body’s fluid homeostasis via blood pressure control [64]. For example, Ang II can increase water drinking by subfornical organ affecting [65]. That is why most hypertension-related animal studies focused on PVN alteration, including inflammation [43] and BBB leakage [61,66].

In comparison, there were few studies about how exercise affects the BBB integrity under the disease condition. For example, exercise may improve the BBB leakage for multiple sclerosis patients [39,40]. The possible mechanism might be that exercise could stimulate circulating angiogenic cells to restore endothelial cell function of the brain blood vessels [67]. However, to the best of our knowledge, we did not find any study about exercise improving hypertension-induced BBB leakage of the hippocampus. Therefore, we will have further studies investigating the detailed mechanism for exercise restoring hypertension-induced BBB leakage.

In this study, we continuously monitored the BBB alteration by blood plasma S100B level measurement [42]. Also, we detected the BBB permeability by IgG immunostaining on the brain section [41]. Interestingly, these two examinations showed inconsistent results in our model. That might be caused by the fact that S100B is not an ideal marker for BBB leakage. For example, a study showed that the level of plasma S100B could be affected by brain injury and several peripheral alterations such as fractures, burns, and even bruises [68]. Moreover, exercise could affect plasma S100B levels as well. A systematic review indicated that exercise could change the plasma S100B to different levels by different exercise protocols [69]. Therefore, the plasma S100B might not be an ideal BBB leakage marker for our study, as the previous study described [70].

In conclusion, we used 2K1C surgery to establish a hypertensive mouse model, and we found that exercise could restore the spatial memory function of this model. Furthermore, we found that running exercise normalized the activated RAS, improving the AHN and BBB leakage in the hippocampus without lowering blood pressure. This study is the first one to investigate how exercise prevents hypertension-induced hippocampal alteration to the best of our knowledge. Further studies are needed for the underlying mechanism to clarify how exercise improves RAS and BBB leakage.

## 4. Materials and Methods

### 4.1. Animals

All experiments were conducted according to the National Institutes of Health Guidelines for animal research (Guide for the Care and Use of Laboratory Animal), and were approved by the Kaohsiung Medical University Institutional Animal Care and Use Committee (Approval number: 108086). The C57BL/6 mice were obtained from the National Laboratory Animal Center (C57BL/6JNarl, Taipei, Taiwan). All of the mice were housed (three mice per cage) in a 12 h light/12 h dark cycle schedule (lights on at 07:00) under constant temperature and humidity (temperature 21 ± 1 °C, humidity 55~70%) with unrestricted access to food and water in the Kaohsiung Medical University Center of Laboratory Animals. We used 8–11 mice per group for the Morris water maze, and we took eight mice among each group for further staining or proteomics study.

### 4.2. Hypertension Animal Model

Nine-week-old C57BL/6 mice underwent the 2K1C surgery to induce hypertension as previously described [41]. Briefly, the mice were anesthetized by intraperitoneal injection of a mixture of tranquilizers (Zoletil 50, 100 mg/kg body weight, Vibrac, Amherst, MA, USA; Rompun, 1 mg/kg body weight, Bayer, Toronto, ON, Canada). The left renal artery was exposed and constricted by inserting a cut-open polyethylene tubing (SP8, outer diameter: 0.5 mm, inner diameter: 0.12 mm, Natsume Seisakusho, Tokyo). Then, a 6-0 silk suture was tied to the polyethylene tubing for position maintenance. The sham group mice received the same 2K1C procedure but without the polyethylene tubing placement. The mice blood pressure was increased by approximately 20% after surgery seven days later and this was maintained for 28 days as previously described [23].

### 4.3. Blood Pressure Measurement

All blood pressure of the mice was monitored by a tail-cuff blood analysis system (BP-2000 system, VisiTech Systems, Apex, NC, USA) one day before the surgery, 21 days after the surgery, and one day after the exercise procedure.

### 4.4. Treadmill Exercise

The American College of sports medicine (ACSM, 2016) recommends that hypertension patients have moderate-intensity aerobic exercise with a frequency of 150 min per week. Another study showed that exercise could regulate the RAS when undertaken with moderate intensity [53]. Therefore, we performed moderate-intensity exercise thirty minutes per day for five days per week. All mice were divided into sedentary and exercise (Ex) groups. The Ex group mice received the moderately intense exercise protocol as previously described [29]. All mice were subjected to five-week exercise training 28-days after the surgery. There was a two day interval between weekly training. On the first week of exercise training, the mice were run on a leveled motor-driven treadmill (Model T510E, Diagnostic and Research Instruments Co., Taoyuan, Taiwan) at 8 m/min speed for 30 min/day for five days. In the second week, the mice were run on the treadmill at 10 m/min speed for 30 min for five days. The mice were run on the treadmill at 12 m/min speed for five days from the third to fifth weeks. The sedentary group was placed on the treadmill for 10 min each day without exercise training. If the mice touched the back of the treadmill, the operator would touch the tail gently to make them run forward.

### 4.5. Morris Water Maze

The Morris water maze was performed in a circular pool (diameter of 120 cm and a wall height of 60 cm) filled with opaque water (diluted milk) at a temperature of 25 ± 2 °C and depth of 35 cm, as previously described [23]. A circular transparent acrylic escape platform (diameter 10 cm) was flooded 1 cm under the water surface during the training trials. The location of the hidden escape platform was constant (target quadrant) in the training trials period (Day one to Day three). The mice were placed in the random quadrant (besides the target quadrant) and allowed to explore for 60 s three times each day. In the probe test (Day four), the mice were placed in the opposite quadrant and allowed to explore for 60 s with the escape platform removed. The swim path was recorded by a CCD camera (C615, Logitech, Lausanne, Switzerland) and used to analyze the time of escape latency (training trial) and the time spent in the target quadrant (probe test). All videos were analyzed by ImageJ (1.53j, National Institutes of Health, Bethesda, MD, USA).

### 4.6. Tissue Collection

One day after the behavior assessment, the mice were anesthetized by intraperitoneal injection of a mixture of tranquilizers (Zoletil 50, 100 mg/kg body weight, Vibrac, Amherst, MA, USA; Rompun, 1 mg/kg body weight, Bayer, Toronto, Canada) and 1 mL of blood was collected from the left heart ventricle. The mice were then perfused with 10 mL cold-PBS buffer (IB3011, Omicsbio, Taipei, Taiwan). After PBS perfusion, the mice brains were quickly removed, dissected into the cortex and hippocampus (left hemispheres, stored at −80 °C), and fixed in 10% neutral formalin (BS chemicals, Kretinga, Lithuania, right hemispheres, held at 4 °C).

### 4.7. Immunohistochemistry (IHC)

The formalin-fixed tissues were dehydrated, paraffin-embedded, and cut into 4-μm thick slides. Before the IHC staining, the brain slides were deparaffinized by xylene, rehydrated through graded ethanol, and antigen retrieval was by 0.1 M citric acid (pH = 6.0, 95 °C for 30 min). The slides were then treated with 2% H_2_O_2_ (Room temperature for 30 min) and blocked with bovine serum albumin (BSA, 3% in PBS/0.5% Triton X-100, room temperature for 1 h). After blocking, the slides were probed with rabbit anti-ki67 antibody (AB9260, 1:300, Millipore, Burlington, MA, USA) to observe the neuronal stem cell of the dentate gyrus; and the neuronal progenitor cell of the dentate gyrus was probed by mouse anti-doublecortin antibody (DCX, C-18, 1:300, Santa Cruz, Dallas, TX, USA). All slides were followed to incubate with appropriate secondary antibodies provided by the IHC kit (Super Sensitive Polymer-HRP IHC Detection System, Biogenex, Fremont, CA, USA) as previously described [71], and the signal was developed by 3, 3’-diaminobenzidine (DAB) substrate. All staining results were analyzed by ImageJ (1.53j, National Institutes of Health, NIH, Bethesda, MD, USA). For Ki67 quantification, only the Ki67^+^ cells in the subgranular zone would be counted (Appendix A.). We removed the molecular layer, hilus, and granule cell signal in the first step. The removed part would fill up with the background color. Next, we generated the 8-bit picture and binary map by signal threshold. Before the particle analysis, we removed the outlier signal according to size and shape discerption. After the count map was generated by particle analysis, we merged the count map with the original picture and counted the actual neural stem cell numbers. Only the cell with an intact soma shape would be counted (Appendix A.).

The BBB leakage was determined as previously described [41]. In short, after the slide blocking, the slides were probed with Goat anti-mouse IgG antibody (BA9200, 1:500, Vector Laboratories, Burlingame, CA, USA) (RT for 16 h). Then, all slides were incubated with avidin-biotin-peroxidase provided by a kit (Vector Vectastain Elite ABC kit, PK-6101, Vector Laboratories, Burlingame, CA, USA) for 30 min. After washing out the solution, the signal was developed by DAB substrate (Vector DAB kit, SK-4100, Vector Laboratories, Burlingame, CA, USA).

### 4.8. Immunoblotting

The frozen mouse brains were homogenized by lysis buffer (50 mM Tris-HCl, 10 mM EDTA, 1% SDS, 0.5% Triton X-100, pH 7.4), containing a protease inhibitor cocktail (Roche, Basel, Switzerland). The homogenate was then centrifuged at 16,000× *g*/4 °C for 90 min, and the supernatant was stored at −80 °C until used. The concentration of the supernatants was determined by the Bio-RAD protein assay kit (#5000006, BioRad, CA, USA) and adjusted to the same concentration (40 μg in 20 μL). The sample was followed mixed with 5× loading dye (1.51 g tris base, 20 mL glycerol, 25 mL ddH_2_O, 4 g/100 mL SDS, 10 mL/100 mL 2-Mercaptoethanol, 0.004 g Bromophenol-blue) and heated at 95 °C for 10 min, then resolved in 4–12% polyacrylamide gels. The resolved samples were transferred to polyvinylidene difluoride membrane (IEVH85R, Millipore, Burlington, MA, USA), blocked with TBST (20 mM Tris/HCl, 0.15 M NaCl, 0.5% Tween-20, pH 7.4) containing 3% nonfat milk for 1 h. The membranes were then incubated with the following primary antibodies at 4 °C overnight: rabbit anti-Angiotensin II (1:1000, A1588-100, Biovision, Milpitas, CA, USA) for angiotensin II, rabbit anti-AGTR1 (1:1000, 25343-1-AP, proteintech, Rosemont, IL, USA) for angiotensin receptor type I, rabbit anti-AGTR2 (1:1000, A00432, Boster Biological Technology, Pleasanton CA, USA) for angiotensin receptor type II, and mouse anti-β-actin (1:2000, 60008-1-lg, proteintech, Rosemont, IL, USA) for protein loading control. After washing, the membranes were probed with appropriate horseradish peroxidase-conjugated secondary antibodies: Ang II, AGTR1, and AGTR2 are goat anti-rabbit IgG HRP (AS014, Abclonal, Woburn, MA, USA); andβ-actin is goat anti-mouse IgG HRP (AS003, Abclonal, Woburn, MA, USA). After incubated with secondary antibodies for 2 h, the signal was developed by chemiluminescence detection (WBLUF0500, Millipore, Burlington, MA, USA). The detection was carried out with an image analysis system (K16C II, Wealtec, NV, USA) and analyzed by ImageJ (1.53j, National Institutes of Health, Bethesda, MD, USA).

### 4.9. ELISA

The collected whole blood was centrifuged at 1500× *g* for 10 min, and the supernatant plasma was gathered for the following plasma Ang II determination. The reagents and procedures were provided from a commercially available Angiotensin II ELISA kit (Ang II, RK02592, ABclonal, Woburn, MA, USA) and an S100B (E03S0042, BlueGene, Shanghai, China). Briefly, the plasma was mixed with the equipment providing diluent (1:1, w:v) and applied to the antibody-coated 96-well plate. The secondary antibody was then incubated for 30 min at room temperature after washing with PBS. The TMB solution was added for 5 min and stopped by a stop reagent for signal development. The signal was determined by a spectrophotometer reader (µQuant, Bio-Tek Instruments, Winooski, VT, USA) at an absorbance wavelength of 450 nm.

### 4.10. Sholl Analysis

We performed a Sholl analysis as previously described for dendritic complexity measurement [23]. Only the neurons with intact soma and connected dendrites from the dentate gyrus would be used for the Sholl analysis (Appendix A). We used a 40× objective lens to catch the dendrite image, then traced with the simple neurite tracer plugin from ImageJ software to label the neurite, as Appendix A shows. The 5-μm increment concentric rings were generated and analyzed by the neuroanatomy > Sholl analysis (From image…) plugin from ImageJ software. Three to five neuron Sholl analysis results from each mouse were averaged as one individual sample size (Appendix A).

### 4.11. TUNEL Assay

The apoptotic cells were detected by a commercial TUNEL assay kit (In Situ Cell Death Detection Kit, POD, ROC-11684817910, Roche, Mannheim, Germany). All procedures were followed according to the manufacturer’s instructions. The 4-μm thick paraffin slides were deparaffinized, rehydrated, and antigen was retrieved, as mentioned. The slides were then permeabilized (0.1% Triton X–100, 0.1% sodium citrate) for 2 min at 4 °C and incubated with the TUNEL reagent for 60 min at 37 °C. After washing the reagent with PBS, the slices would react with the kit-provided POD solution for 30 min at 37 °C. The apoptotic cell signal was developed by the DAB substrate.

### 4.12. Statistical Analysis

All data were plotted and analyzed by GraphPad Prism 8.2.1(GraphPad Software Inc., San Diego, CA, USA). The Morris water maze studies were analyzed by repeated two-way ANOVA followed by Holm-Sidak’s multiple comparisons test. Immunohistochemistry and TUNEL assays were analyzed by using one-way ANOVA followed by Holm-Sidak’s multiple comparisons. Immunoblotting and ELISA assay were analyzed by using one-way ANOVA followed by Holm-Sidak’s multiple comparisons. All results have been shown as means ± S.D. In all statistical comparisons, *p* < 0.05 was considered to be a significant difference.

## Figures and Tables

**Figure 1 ijms-23-05531-f001:**
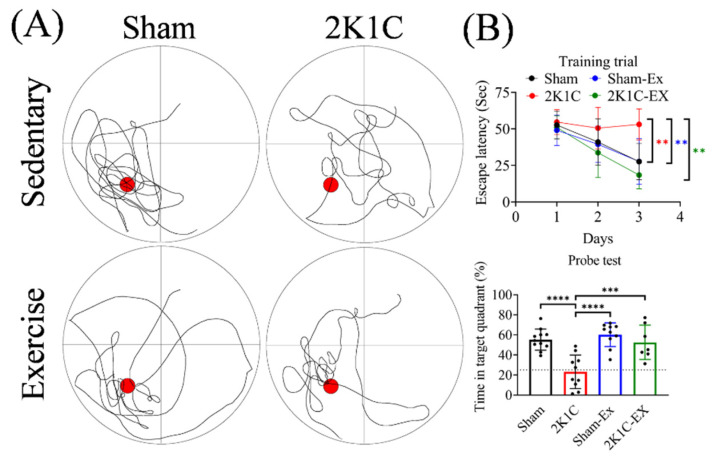
Exercise improved the 2K1C-hypertension-induced spatial memory impairment. (**A**) Maze schematic with swimming trajectory for different groups of mice. Red circle: removed escape platform. (**B**) Upper: Acquisition escape latency to find a hidden platform by day in the training trials. Lower: The time in target quadrant for hidden platform removed probe test. The averaged data and s.d. are plotted. The statistical analysis was performed by one-way ANOVA and Holm-Sidak’s multiple comparisons. The sample sizes for sham, 2K1C, Sham-Ex, and 2K1C-EX are 11, 10, 10, and seven, respectively. **: *p* < 0.01, ***: *p* < 0.001, ****: *p* < 0.0001.

**Figure 2 ijms-23-05531-f002:**
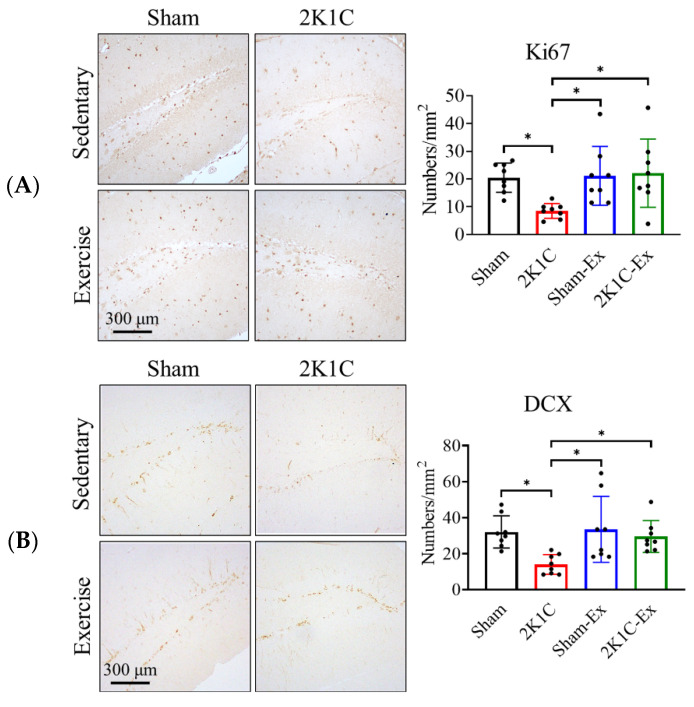
Exercise prevents the 2K1C-hypertension-induced AHN impairment. The immunohistochemistry micrographs for (**A**) Ki67^+^ neuronal stem cells and (**B**) DCX^+^ neuronal progenitor cells. The averaged data and s.d. are plotted. The statistical analysis was performed by one-way ANOVA and Holm-Sidak’s multiple comparisons. The sample sizes are seven for each group. *: *p* < 0.05. Scale bar, 300 μm.

**Figure 3 ijms-23-05531-f003:**
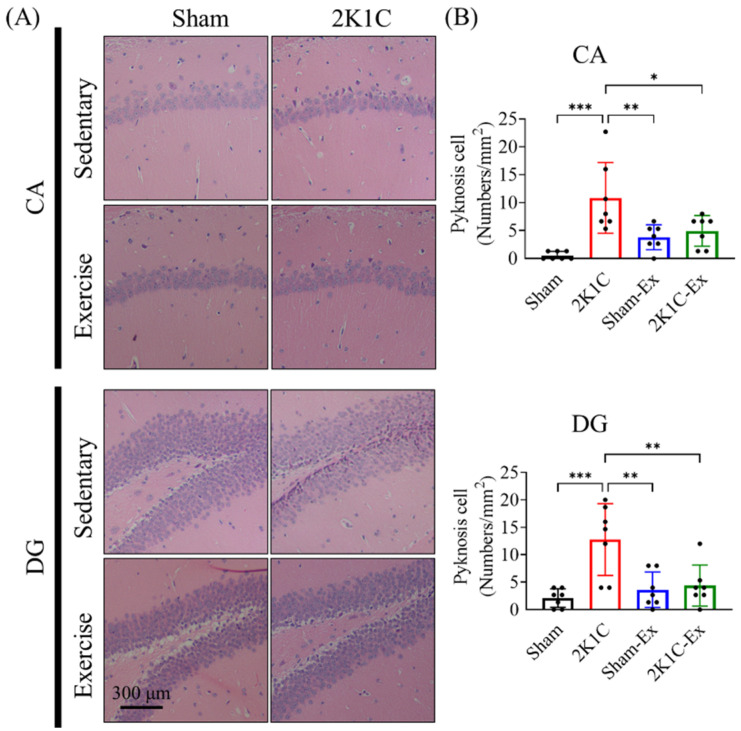
Exercise prevents the 2K1C-hypertension-induced hippocampus neuronal cell pyknosis. (**A**) The H&E staining micrographs for CA1 (CA) and dentate gyrus (DG) of hippocampal. (**B**) The quantitative result for pyknosis cell density in the CA and DG. The averaged data and s.d. are plotted. The statistical analysis was performed by one-way ANOVA and Holm-Sidak’s multiple comparisons. The sample sizes are seven for each group. *: *p* < 0.05 **: *p* < 0.01, ***: *p* < 0.001. Scale bar, 300 μm.

**Figure 4 ijms-23-05531-f004:**
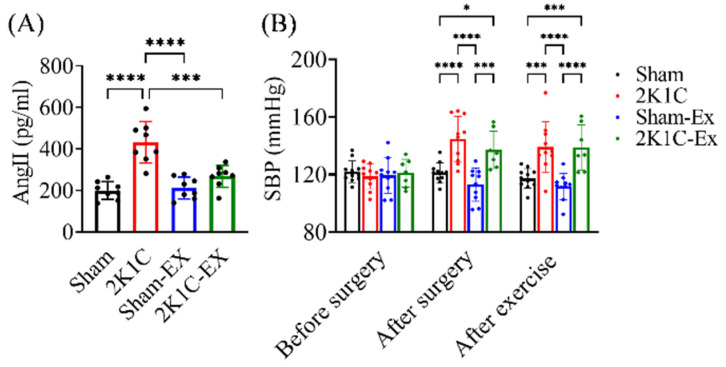
Exercise restores the 2K1C-hypertensive mouse model plasma angiotensin II level without decreasing the blood pressure. (**A**) The plasma Ang II level was measured by ELISA. The statistical analysis was performed by one-way ANOVA and Holm-Sidak’s multiple comparisons. The averaged data and s.d. are plotted. The sample sizes are seven for each group. ***: *p* < 0.001, ****: *p* < 0.0001. (**B**) The systolic blood pressure was measured before and after surgery and after exercise training. The statistical analysis was performed by two-way ANOVA and Holm-Sidak’s multiple comparisons. The averaged data and s.d. are plotted. The sample sizes for sham, 2K1C, Sham-Ex, and 2K1C-EX are 11, 10, 10, and 7, respectively. *: *p* < 0.05, ***: *p* < 0.001, ****: *p* < 0.0001.

**Figure 5 ijms-23-05531-f005:**
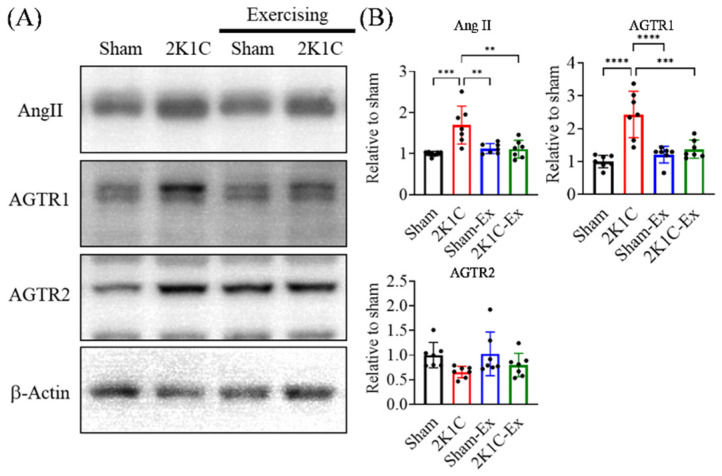
Exercise downregulates the 2K1C-hypertensive mouse model hippocampus RAS. (**A**) Representative immunoblots for each group of hippocampus tissue of the mice. (**B**) The quantitative result for protein level of Ang II, Angiotensinogen type I receptor (AGTR1), and Angiotensinogen type II receptor (AGTR2). The statistical analysis was performed by one-way ANOVA and Holm-Sidak’s multiple comparisons. The averaged data and s.d. are plotted. The sample sizes are seven for each group. **: *p* < 0.01, ***: *p* < 0.001, ****: *p* < 0.0001.

**Figure 6 ijms-23-05531-f006:**
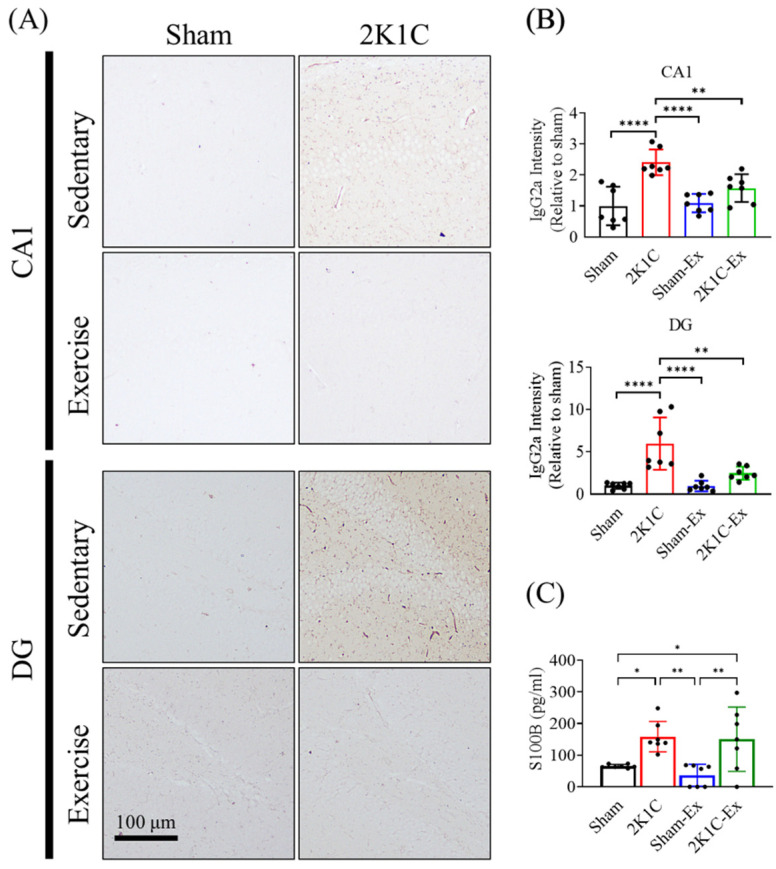
Exercise prevents the 2K1C-hypertensive mouse model hippocampus BBB leakage. (**A**) The immunohistochemistry micrographs for the mouse-IgG2a staining in the hippocampus CA1 and dentate gyrus (DG). (**B**) The quantitative result for IgG2a antibody level in the hippocampus CA1 and DG. (**C**) The plasma S100B level was measured by ELISA. The statistical analysis was performed by one-way ANOVA and Holm-Sidak’s multiple comparisons. The averaged data and s.d. are plotted. The sample sizes are seven for each group. *: *p* < 0.05, **: *p* < 0.01, ****: *p* < 0.0001. Scale bar, 100 μm.

## Data Availability

The data that supported this study are available from the corresponding author upon reasonable request.

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
