# Peer review of "Exercise Normalized the Hippocampal Renin-Angiotensin System and Restored Spatial Memory Function, Neurogenesis, and Blood-Brain Barrier Permeability in the 2K1C-Hypertensive Mouse"

_ijms, 2022, doi:10.3390/ijms23105531_

Round 1

Reviewer 1 Report

Manuscript ID: ijms-1701550

Manuscript Title: Exercise Normalized the Hippocampal Renin-Angiotensin System and Restored Spatial Memory Function, Neurogenesis, and Blood-Brain Barrier Permeability in the 2K1C-Hypertensive Mouse

Authors investigated the effects of treadmill exercise on hypertension-induced impairments in spatial memory function, neurogenesis, and BBB permeability based on behavioral, morphological, and biochemical approaches. Results are interesting, but there are some problems to analyze the data and in specificity of Ki67 antibody.

Most Ki67-positive cells were found in the subgranular zone of dentate gyrus because it is a marker for active cell cycle. Authors showed microphotographs of Ki67-stained cells in Fig. 2A, but the specificity of Ki67 is doubtful.

In Fig. 2B, authors demonstrated the number of DCX-positive neuronal progenitor cells. How did the authors count the number of cells? In addition, I suggest that authors show the dendritic complexity with Sholl analysis because the dendritic complexity is significantly different among groups.

In Fig. 3. authors observed the pyknosis cells in the subgranular zone of dentate gyrus. I suggest that authors conduct TUNEL staining to confirm the neuronal damage in the subgranular zone.

Authors described that they monitored the blood pressure one day before the surgery, 21 days 315 after the surgery, and one day after the exercise procedure. However, I could not observe any blood pressure data. Authors should demonstrate this and discuss the correlation with blood pressure and BBB leakage because blood pressure is one of important factors in BBB integrity.

Authors should change Zoleti into Zoletil in line 307.

Author Response

Thanks for the reviewer time for reading. We apologize for the delayed submission due to the quarantine.

Reviewer 2 Report

In this manuscript (ID# ijms-1701550), entitled “Exercise Normalized the Hippocampal Renin-Angiotensin System and Restored Spatial Memory Function, Neurogenesis, and Blood-Brain Barrier Permeability in the 2K1C-Hypertensive Mouse”, authors, Chang et al, studied the effect of exercise on renin-angiotensin system (RAS), memory function, and blood brain barrier (BBB) permeability using  the 2-kidney 1-clip mouse hypertensive model. The results demonstrated that exercise inhibited RAS and improved memory and BBB function. However, this study is too superficial, without investigation on the underlying mechanisms. Several major concerns are provided in the following paragraphs:

  1. What is the novelty in the current study? The authors mention “we did not find any study about exercise improving hypertension-induced BBB leakage” in page 8, lines 272-273. Actually, it has been well studied that exercise improves hypertension-related BBB leakage, e.g. Front Physilo 2017, 8:1048, doi: 10.3389/fphys.2017.01048; Am J Physiol Regul Integr Comp Physiol 2021, 321: R732, doi: 10.1152/ajpregu.00154.2020.
  2. The blood pressure was measured using an old tail-cuff method, which is not reliable. A new technique, radiotelemetry, is available. This inaccurate technique may contribute the phenomena that exercise did not decrease systolic blood pressure, observed in this study.
  3. In Fig 6, the BBB leakage was detected by immunohistochemistry using IgG2a and S100B antibodies in the brain tissue. The production of IgG2a and S100B could be also increased by inflammation in the brain.
  4. In Fig 5, Ang II, AT1R, and AT2R was examined. Other members in the renin-angiotensin system should also be detected in the brain, such as ACE, renin, in order to understand why Ang II was elevated.

Author Response

We thank for the reviewer time for reading, and we apologize for the delayed submission due to the quarantine.

Round 2

Reviewer 1 Report

Although I rejected the manuscript on 1st round of review, authors thoroughly edited the manuscript. In present state, the most of concerns were resolved and I have no further comment. 

Author Response

Thanks for the reviewer's valuable suggestion. We hope the edited manuscript can meet the journal's high standard expect.

Reviewer 2 Report

This manuscript was not improved significantly.

Author Response

We appreciate the reviewer's time reading and providing essential suggestions to help us revise the manuscript. The database analysis result we offered in this round is still not enough for publication; therefore, we do not intend to add it to the manuscript. 
